# Supramolecular Assemblies in Mn(II) and Zn(II) Metal–Organic Compounds Involving Phenanthroline and Benzoate: Experimental and Theoretical Studies

**Mridul Boro** [1], **Subham Banik** [1], **Rosa M. Gomila** [2], **Antonio Frontera** [2,*], **Miquel Barcelo-Oliver** [2] and **Manjit K. Bhattacharyya** [1,*]

1    Department of Chemistry, Cotton University, Guwahati 781001, Assam, India; boromridul8@gmail.com (M.B.); chm2391001_subham@cottonuniversity.ac.in (S.B.)
2    Departament de Química, Universitat de les Illes Balears, Crta de Valldemossa km 7.7, 07122 Palma de Mallorca, Spain; rosa.gomila@uib.es (R.M.G.); miquel.barcelo@uib.es (M.B.-O.)
*    Correspondence: toni.frontera@uib.es (A.F.); manjit.bhattacharyya@cottonuniversity.ac.in (M.K.B.)

**Abstract:** Two new Mn(II) and Zn(II) metal–organic compounds of 1,10-phenanthroline and methyl benzoates *viz*. [Mn(phen)$_2$Cl$_2$]2-ClBzH (**1**) and [Zn(4-MeBz)$_2$(2-AmPy)$_2$] (**2**) (where 4-MeBz = 4-methylbenzoate, 2-AmPy = 2-aminopyridine, phen = 1,10-phenanthroline, 2-ClBzH = 2-chlorobenzoic acid) were synthesized and characterized using elemental analysis, TGA, spectroscopic (FTIR, electronic) and single crystal X-ray diffraction techniques. The crystal structure analysis of the compounds revealed the presence of various non-covalent interactions, which provides stability to the crystal structures. The crystal structure analysis of compound **1** revealed the formation of a supramolecular dimer of *2-ClBzH* enclathrate within the hexameric host cavity formed by the neighboring monomeric units. Compound **2** is a mononuclear compound of Zn(II) where flexible binding topologies of *4-CH$_3$Bz* are observed with the metal center. Moreover, various non-covalent interactions, such as lp(O)-π, lp(Cl)-π, C–H⋯Cl, π-stacking interactions as well as N–H⋯O, C–H⋯O and C–H⋯π hydrogen bonding interactions, are found to be involved in plateauing the molecular self-association of the compounds. The remarkable enclathration of the H-bonded 2-ClBzH dimer into a supramolecular cavity formed by two [Mn(phen)$_2$Cl$_2$] complexes were further studied theoretically using density functional theory (DFT) calculations, the non-covalent interaction (NCI) plot index and quantum theory of atoms in molecules (QTAIM) computational tools. Synergistic effects were also analyzed using molecular electrostatic potential (MEP) surface analysis.

**Keywords:** metal–organic compounds; supramolecular dimer; self-association; DFT; QTAIM; MEP

## 1. Introduction

Molecular self-assembly, especially based on inorganic metal ions and organic ligands, stands out as a highly efficient and widely employed strategy for constructing molecular architectures. This methodology holds significant relevance in numerous fields, including catalysis, sensors, semiconductor devices, luminescent materials, and in biology [1–5]. Its broad application stems from the fascinating structural topologies it engenders [6–9]. A key point for the synthesis of desired network architectures requires mutual adaptation between geometries of metal ions and the selection of proper ligands as building blocks [10]. Also, the structural topologies of coordination frameworks are profoundly influenced by factors such as the coordination geometry or size of metal ions, guest molecules, and counterions, as well as a variety of experimental conditions like solvent choice, the metal-to-ligand ratio, reaction duration, pH levels, etc. [11–15].

Scientific communities have been delving into the intricate world of supramolecular chemistry. This recent exploration has focused on unraveling various non-covalent interactions, including hydrogen bonding (HB), stacking, and charge transfer interactions, with

particular emphasis on their implications in crystal engineering [16,17]. Also, the compounds rely on the precise coordination of molecular self-assembly, which is controlled by weak non-covalent interactions viz. anion-π, cation-π, π-stacking, C–H/π, σ/π-hole, lone-pair/π, halogen bonding, etc., which through their collective strength, directional control, and synergistic effects, are integral for maintaining compound stability [18–20]. The intriguing phenomenon of cooperative reciprocity among π-stacking interactions has captured the attention of researchers, especially within the framework of crystal engineering [21,22].

The art of designing coordination complexes utilizing N-donor heterocyclic organic compounds in collaboration with aromatic carboxylates still holds the attention of researchers [23,24]. 4-Mebz has attracted great interest owing to its two interesting structural features. Firstly, its multiple bridging moieties allow for diverse bonding modes with transitioning metal centers, leading to a plethora of structural arrangements [25]. In the second place, it can act not only as hydrogen bond donors but also as acceptors due to the existence of protonated and/or deprotonated carboxyl groups [26]. 1,10-phenanthroline (*phen*), a heterocyclic bidentate N-donor, competently generates stable coordination compounds with various transition metals on account of its chelating nature [27]. It thereby holds an exclusive place as a primary material in coordination chemistry [28–33]. The presence of electron-deficient aromatic systems in phen makes it an excellent electron acceptor capable of stabilizing metal complexes via various unconventional non-covalent interactions [34]. Pyridine-based donors differing in substituents and stereochemistry have also given rise to immense research interests in coordination chemistry due to their potential applications in diverse fields [35,36]. Amongst the pyridine derivatives, 2-Aminopyridine is used for the synthesis of pharmacologically active heterocyclic molecules [37]. 2-Aminopyridine is a good neutral donor of mono and bidentate nature, and its coordination properties can be easily altered by substitution with either electron-donating or electron-withdrawing groups, leading to the improvement of desired pharmacological activities [38–45]. The coordination compounds of zinc involving benzoate and substituted benzoate derivatives have been reported to possess interesting structural topologies [46,47]. Manganese complexes are found to show moderate-to-strong inhibition against different human cancer cells in vitro [48].

The inclusion of self-assembled guests in host cavities within supramolecular architectures typically relies on both molecular associations and the size of the molecules involved [49]. Non-covalent bonding in supramolecular inclusion complexes, apart from demonstrating precise three-dimensional architectures, also confers intrinsic reversibility and adaptivity, enabling dynamic responsiveness to external stimuli [50–52]. The host–guest molecules' high selectivity fosters dynamic interactions within molecular self-assemblies, paving the way for the development of supramolecular soft biomaterials with intricate structures and programmable functions [53–55].

Herein, we describe the synthesis and crystal structures of two newly synthesized Mn(II) and Zn(II) metal–organic compounds incorporating phen and 4-$CH_3$Bz. Characterization was performed using FT-IR and electronic spectroscopy, as well as elemental and thermo-gravimetric (TG) analysis, aiming to elucidate the role of non-covalent interactions in the molecular self-assembly of mononuclear coordination compounds. Through the utilization of single-crystal X-ray diffraction, we revealed the crystal structures of the compounds. Additionally, we investigated a range of non-covalent interactions contributing to the molecular association of these compounds. The crystal structure analysis of compound **1** demonstrated the dimerization of 2-ClBzH moieties within the lattice, followed by their enclathration within the hexameric supramolecular host cavity formed by the orderly assembly of monomeric units. Similarly, the crystal structure analysis of compound **2** revealed the dual coordination mode of 4-$CH_3$Bz with an identical metal center, elucidating their involvement in the self-aggregation of individual units and leading to the formation of unique supramolecular architectures. Moreover, non-covalent interactions involving lp(O)-π, lp(Cl)-π, C–H⋯Cl, π-stacking interactions as well as N–H⋯O, C–H⋯O and C–H⋯π hydrogen bonding interactions are involved in stabilizing the molecular self-association of

the compounds. The theoretical study of the enclathration of the H-bonded 2-ClBzH dimer within a supramolecular cavity created by two [Mn(phen)₂Cl₂] complexes was analyzed. This investigation utilized density functional theory (DFT) calculations, the non-covalent interaction (NCI) plot index, and the quantum theory of atoms in molecules (QTAIM) as computational methodologies. Additionally, molecular electrostatic potential (MEP) surface analysis was employed to examine synergistic effects.

## 2. Results and Discussion

### 2.1. Syntheses and General Aspects

[Mn(phen)₂Cl₂]2-ClBzH (**1**) has been synthesized by reacting one equivalent of MnCl₂·4H₂O, one equivalent of 2-ClBzH and two equivalents of phen at room temperature in a water medium. Similarly, [Zn(4-MeBz)₂(2-AmPy)₂] (**2**) was prepared by the reaction between one equivalent of ZnCl₂, two equivalents of Na-4-MeBzH, and two equivalents of 2-AmPy at room temperature in a water medium. The compounds are fairly soluble in water as well as in common organic solvents. Compound **1** showed a room temperature (298 K) $\mu_{eff}$ value of 5.89 BM, which suggested the presence of five unpaired electrons in the Mn(II) center of the distorted octahedral coordination sphere of **1** [56,57].

### 2.2. Crystal Structure Analysis

The molecular structure of compound **1** is shown in Figure 1. Selected bond lengths and bond angles around the Mn(II) centers are summarized in Table S1. Compound **1** crystallizes in the triclinic crystal system with a $P\bar{1}$ space group. As shown in Figure 1, compound **1** is a mononuclear compound of Mn(II), which is hexa-coordinated with two bidentate phen moieties and two monodentate chloride ions. In addition, one uncoordinated 2-ClBzH moiety is also present in the crystal lattice. The coordination geometry around the Mn1 center is a distorted octahedron, where the axial sites are occupied by N10A of phen and Cl1 atoms; on the other hand, the equatorial sites are occupied by N10B, N1B, and N1A from phen moieties and Cl2 atoms. The four equatorial atoms viz. N10B, N1B, N1A, Cl2 are distorted from the mean equatorial plane with the mean r.m.s. deviation of 0.1712 Å. The dihedral angle between the two phen moieties was found to be 85.78°. The average Mn–N and Mn–Cl bond lengths were almost consistent with the previously reported Mn(II) complexes [58,59].

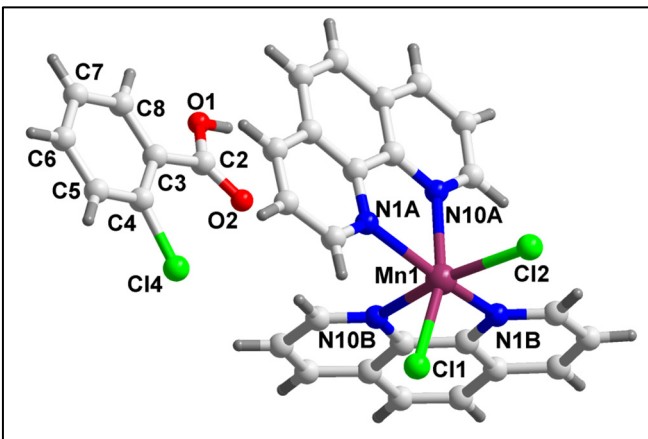

**Figure 1.** Molecular structure of [Mn(phen)₂Cl₂]2-ClBzH (**1**).

The monomeric units of compound **1** are interconnected via weak C–H···Cl hydrogen bonding and aromatic π-stacking interactions, which are responsible for the formation of the 1D supramolecular chain of the compound (Figure 2). The Cl1 atom is involved in two C–H···Cl hydrogen bonding interactions with the –CH moieties of phenwith C8B–H8B···Cl1 and C4B–H4B···Cl1 distances of 2.75(1) and 2.81(1) Å, respectively. Moreover, aromatic π-stacking interactions were also observed between the aromatic rings of phen

(N1B, C2B, C3B, C4B, C12B, and C11B) and (N1B, C2B, C3B, C4B, C12B, and C11B) with a centroid–centroid separation of 3.49(5) Å. The normal ring and the vector between the two ring centroids formed an angle (slipped angle) of about 22.28(8)°, which is close to the slipped π-stacking interactions reported in the literature [60].

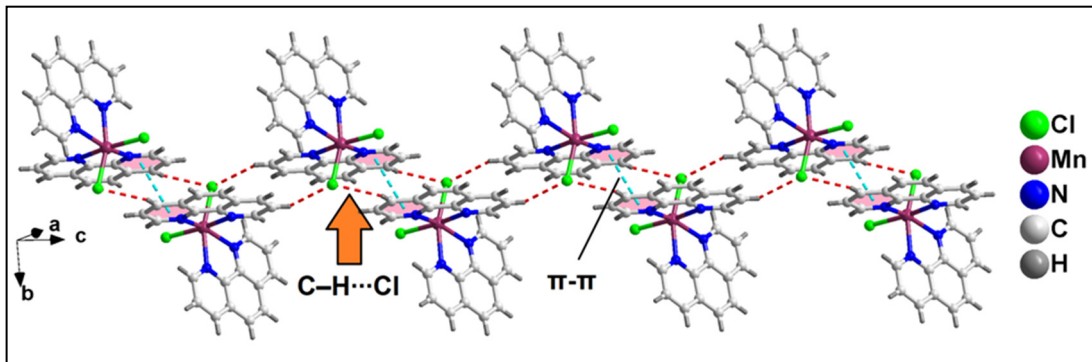

**Figure 2.** The 1D supramolecular chain of compound **1** along the crystallographic c axis assisted by C–H···Cl and π-stacking interactions.

The uncoordinated 2-ClBzH molecules of compound **1** formed a hydrogen-bonded supramolecular dimer assisted by strong O–H···O hydrogen bonding interactions (Figure 3). O–H···O hydrogen bonding interactions between the two 2-ClBzH moieties were observed with an O1–H1···O2 distance of 1.79(3) Å (Table 1). The supramolecular ring motif formed in the cyclic supramolecular dimer was expressed using Etter's graph-set notation [61] viz. $R_2^2(8)$.

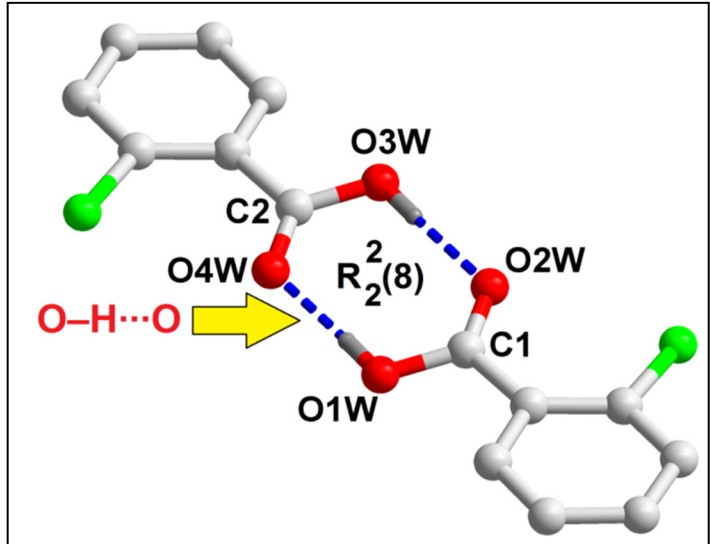

**Figure 3.** Formation of supramolecular dimer of 2-ClBzH in compound **1** assisted by O–H···O hydrogen bonding interactions. Aromatic hydrogen atoms have been omitted for clarity.

The supramolecular dimer was enclathrated within the supramolecular host cavity, formed by six different monomeric units and assisted by the weak C–H···Cl contacts (Figure 4). In addition, O-π contacts were observed involving the pyridine rings of the phen moieties with O1-Cg (Cg is the centroid of the ring formed by the atoms N10B, C9B, C8B, C7B, C14B, and C13B) and O2-Cg distances of 3.18(3) Å and 3.91(3) Å, respectively. Cl-π contacts were also observed involving the pyridine rings of the phen moieties with a Cl4-Cg$_1$ (Cg$_1$ is the centroid of the ring formed by the atoms C12A, C11A, N1A, C2A, C3A, and C4A) distance of 3.92(2) Å. These enclathrated dimers of 2-ClBzH propagated

along the crystallographic *bc* plane to stabilize the layered architecture of the compound (Figure 5).

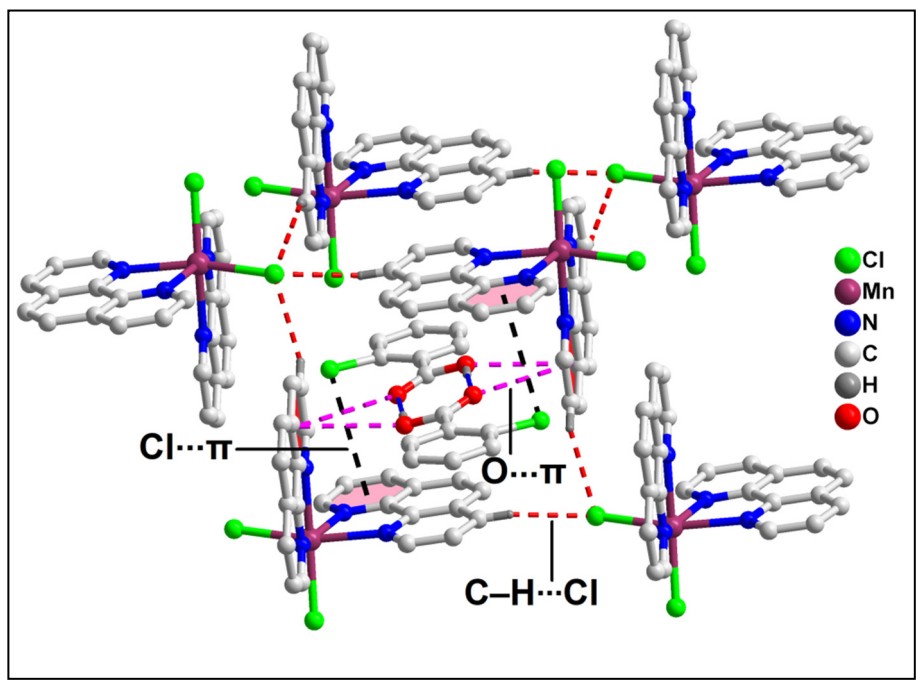

**Figure 4.** Enclathration of the hydrogen bonded dimer of 2-ClBzH inside the supramolecular hexameric host cavity of **1**.

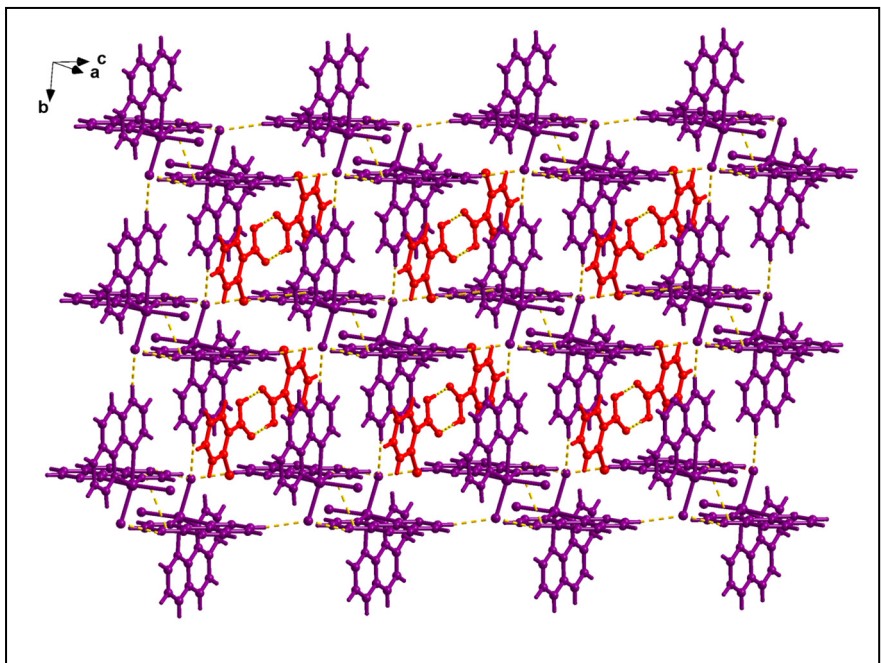

**Figure 5.** Layered assembly of compound **1** along the crystallographic *bc* plane.

Further analysis revealed that C–H···Cl hydrogen bonding interactions play a crucial role in the propagation of the supramolecular 1D chains to form the layered architecture. Along the crystallographic *ab* plane, two types of C–H···Cl hydrogen bonding interactions were observed between the neighboring monomeric units (Figure 6). The Cl2 atom was involved in a C–H···Cl interaction with the pyridyl ring of phen moiety, with a C7B–

H7B···Cl2 distance of 2.92(1) Å; the Cl1 atom was also involved in a C–H···Cl interaction with the pyridyl ring of phen moiety, with a C4B–H4B···Cl1 distance of 2.81(1) Å.

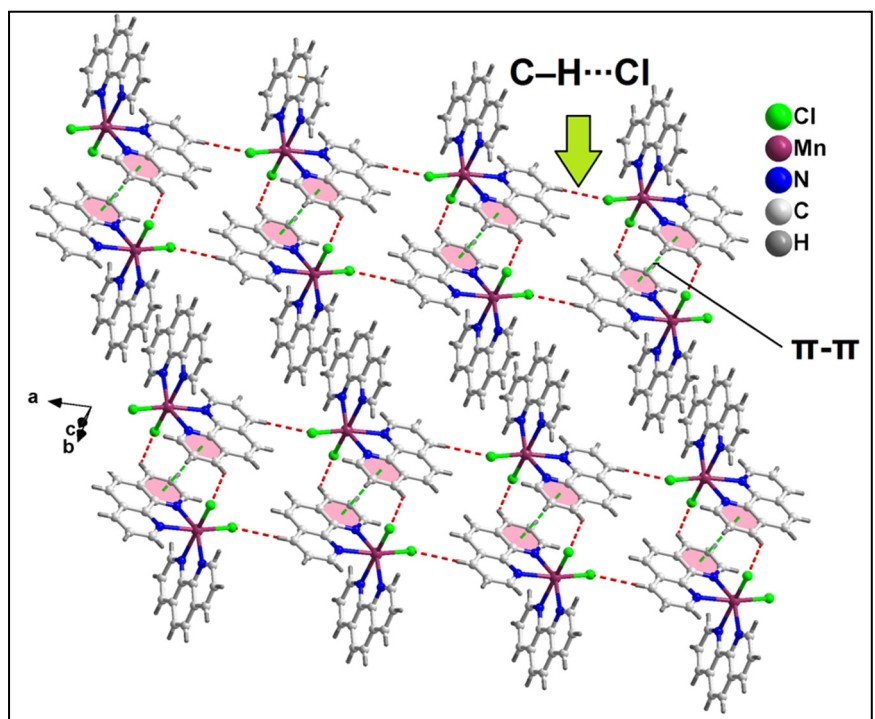

**Figure 6.** Layered assembly of compound **1** along the crystallographic *ab* plane.

**Table 1.** Selected hydrogen bond distances (Å) and angles (deg.) for compound **1**.

| D–H···A | d(D···A) | d(H···A) | <(DHA) |
|---------|----------|----------|--------|
| C7B–H7B···Cl2 | 3.701(4) | 2.92(1) | 139.3(3) |
| O1–H1···O2 | 2.635(5) | 1.79(3) | 177.2(3) |
| C8B–H8B···Cl1 | 3.566(6) | 2.75(1) | 144.3(3) |
| C4B–H4B···Cl1 | 3.502(5) | 2.81(1) | 130.4(3) |

The molecular structure of compound **2** is shown in Figure 7. Selected bond lengths and bond angles around the Zn(II) centers are summarized in Table S1. Compound **2** crystallized in the monoclinic crystal system with a Cc space group. As shown in Figure 7, compound **2** comprised a mononuclear Zn(II) metal center. The Zn1 center in compound **2** was penta-coordinated with two monodentate 2-AmPy moieties, one monodentate 4-MeBz moiety, and one bidentate 4-MeBz moiety. The coordination geometry around the Zn1 center is distorted square pyramidal (evidenced by the value of the trigonality index $\tau = 0.1003$) where the axial site was occupied by the N3 atom, whereas the equatorial sites are occupied by the O2, O3, and O1 atoms from two 4-MeBz moieties and the N1 atom from one 2-AmPy moiety. The equatorial atoms viz. O1, O2, O3, and N1 of the Zn1 center were distorted from the mean equatorial plane with a mean r.m.s. deviation of 0.1120 Å. The average Zn–O and Zn–N bond lengths are almost consistent with the previously reported Zn(II) complexes [62].

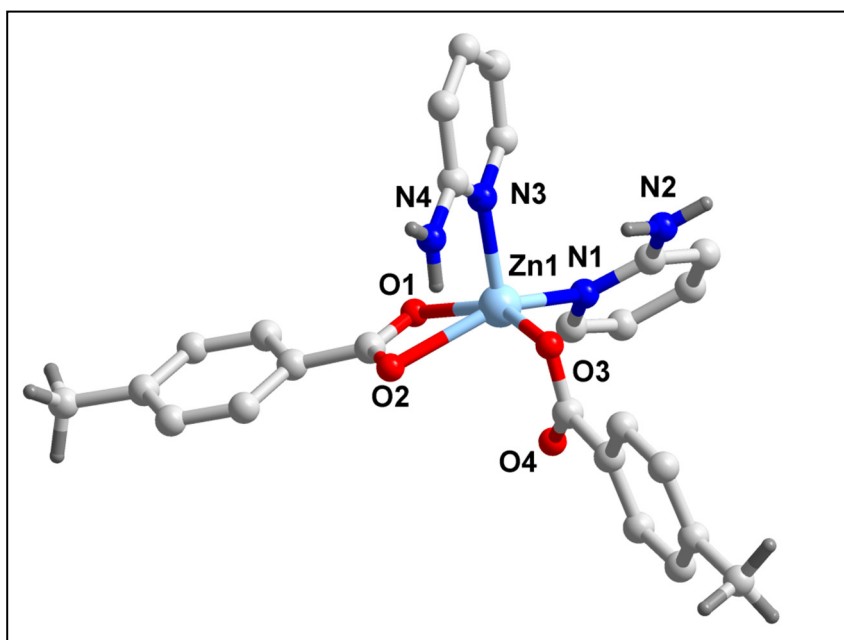

**Figure 7.** Molecular structure of [Zn(4-MeBz)$_2$(2-AmPy)$_2$] (**2**). Aromatic hydrogen atoms are omitted for clarity.

The neighboring complex moieties in compound **2** are interconnected via N–H···O and C–H···O hydrogen bonding interactions with an N2–H2B···O2 distance of 2.03(2) Å and C19–H19···O2, C20–H20···O4 distances of 2.80(3) Å and 2.58(2) Å, respectively (Table 2), to form the 1D supramolecular chain along the crystallographic a axis (Figure 8). Moreover, C–H···π interactions were also observed in the 1D supramolecular chain involving the –CH moieties of 4-MeBz and 2-AmPy and ring centroid of 4-MeBz with H6···Cg$_2$, C6···Cg$_2$ (Cg$_2$ is the ring centroid defined by the atoms C10-C15) and H23···Cg$_2$, C23···Cg$_2$ distances of 2.73(3), 3.66(3) and 2.63(3), 3.52(3) Å, respectively, to provide an extra ballast to the propagation of the 1D chain along the crystallographic a axis. The C6–H6···Cg$_2$ and C23–H23···Cg$_2$ bond angles observed were 165.60(2)° and 156.18(2)°, respectively.

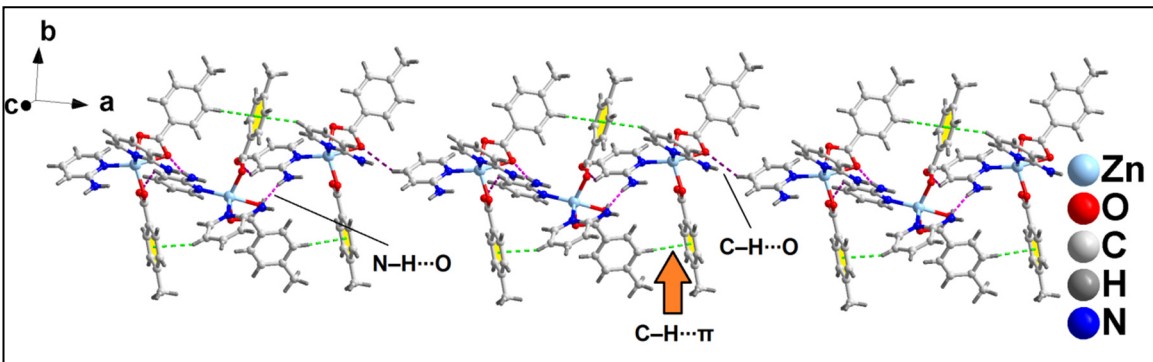

**Figure 8.** The 1D supramolecular chain of compound **2** along the crystallographic a axis assisted by C–H···O, N–H···O hydrogen bonding and non-covalent C–H···π interactions.

The 1D supramolecular chains were interconnected via C–H···O hydrogen bonding interactions to form the layered assembly along the crystallographic *ab* plane (Figure 9). The C–H···O hydrogen bonding interactions were formed between the –C16H16B moiety of 4-MeBz and the O1 atom of another 4-MeBz adjacent monomeric chain with a C16–H16B···O1 distance of 2.68(3) Å.

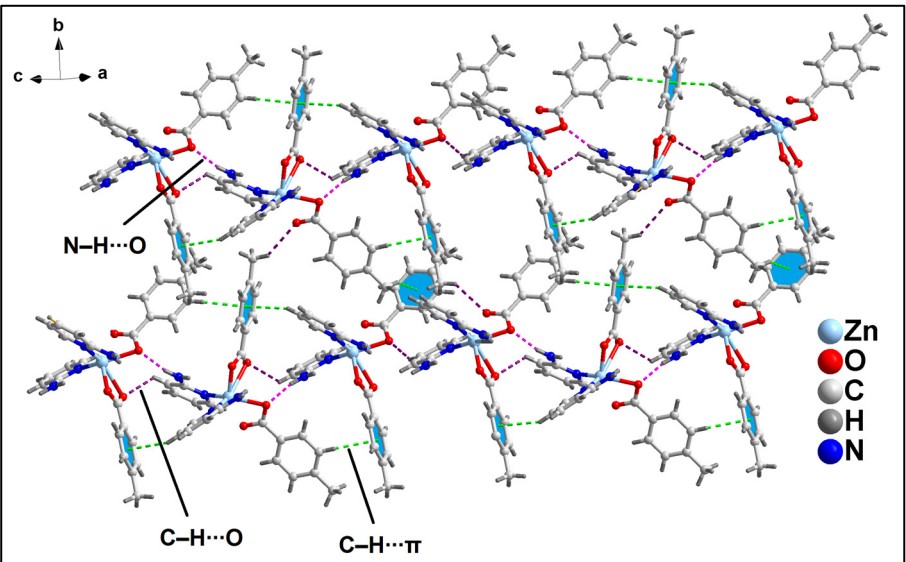

**Figure 9.** Layered assembly of compound **2** along the crystallographic *ab* plane.

Further analysis revealed the presence of C–H···π and C–H···O hydrogen bonding interactions that played a pivotal role in the formation of the layered assembly of compound **2** along the crystallographic ac plane (Figure 10). C–H···O hydrogen bonding interaction was formed between the –CH moiety and O2 atom of 4-MeBz with a C19–H19···O2 distance of 2.80(3) Å. The C–H···π interaction was also observed between the –C24H24 moiety of 2-Ampy and aromatic ring of 4-MeBz with H24···Cg$_3$ and C24···Cg$_3$ (Cg$_3$ is the ring centroid defined by the atoms C2-C7) distances of 2.62(3) and 3.54(3) Å, respectively. The C24–H24···Cg$_3$ bond angle observed was 160.97(2)°.

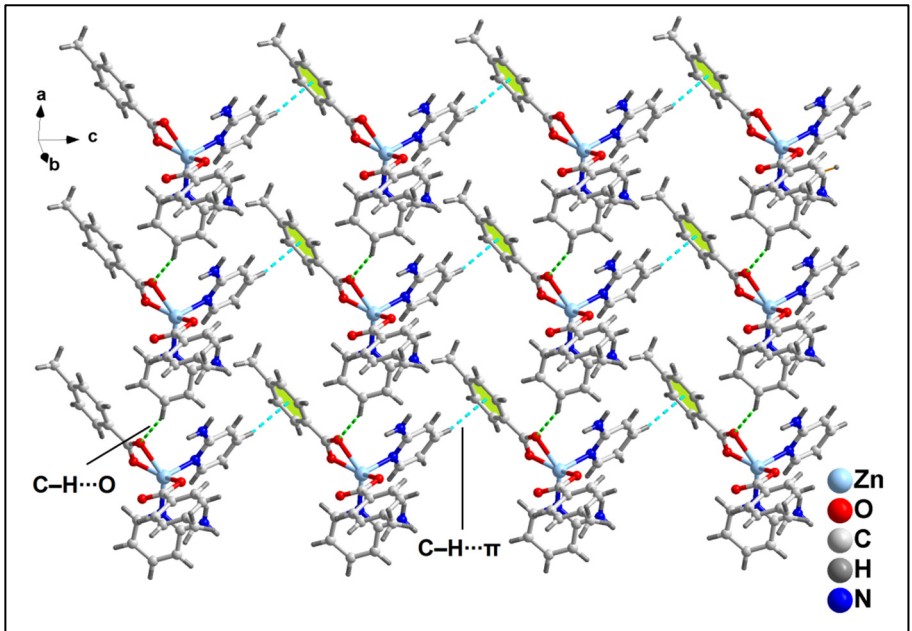

**Figure 10.** Layered assembly of compound **2** along the crystallographic *ac* plane.

**Table 2.** Selected hydrogen bond distances (Å) and angles (deg.) for compound **2**.

| D–H$\cdots$A | d(D$\cdots$A) | d(H$\cdots$A) | <(DHA) |
|---|---|---|---|
| N2–H2B$\cdots$O2 | 2.906(4) | 2.03(2) | 168.435(2) |
| C19–H19$\cdots$O2 | 3.623(5) | 2.80(3) | 145.031(2) |
| C20–H20$\cdots$O4 | 3.197(4) | 2.58(2) | 122.597(2) |
| C16–H16B$\cdots$O1 | 3.522(6) | 2.68(3) | 143.144(3) |

*2.3. Spectral Studies*

2.3.1. FT-IR Spectroscopy

The FT-IR spectra of compounds **1** and **2** were recorded in the region 4000–500 cm$^{-1}$ (Figure S1). The comparatively broad absorption bands in the FT-IR spectra of compound **1** at around 3411 cm$^{-1}$ could be attributed to the O-H stretching vibrations of the carboxyl group of 2-ClBzH moiety present in the crystal lattice [63–65]. The ring stretching vibrations for 2-AmPy ligands were shifted to lower wave numbers (1565, 1453, 1272 cm$^{-1}$) in the FT-IR spectrum of compound **2,** suggesting the coordination of 2-AmPy with a Zn(II) center via a pyridine ring N-atom [66,67]. The wagging vibrations of the pyridine rings were observed at 666 and 695 cm$^{-1}$ [68]. The bands at 1610 and 1496 cm$^{-1}$ in the FT-IR spectrum of **2** could be attributed to the asymmetric $\nu_{as}$(COO) and symmetric $\nu_s$(COO) stretching vibrations of the carboxylate moiety of coordinated 4-MeBz. The difference between the asymmetric and symmetric stretching vibrations of the carboxylate moieties ($\Delta$ < 200 cm$^{-1}$) indicated the bidentate coordination of carboxylate to the respective metal center in compound **2** [69,70]. The absence of any bands near 1710 cm$^{-1}$ in compound **2** indicated the deprotonation of carboxylate groups in the compound [71]. Weak absorptions observed at around 2730–3070 cm$^{-1}$ can be attributed to the $\nu$(C-H) vibrations of the 4-Mebz moieties [72]. In compound **1**, the coordination of phen to the metal centers was confirmed via the shifting of IR frequencies for $\delta$(C-H) vibrations of phen [73,74]. The bands at around 1420 and 1151 cm$^{-1}$ for compound **1** could be attributed to the $\nu$(C=C) and $\nu$(C=N) vibrations of coordinated phen [75].

2.3.2. Electronic Spectroscopy

The electronic spectra of the compounds were recorded in both solid and aqueous phases (Figures S2 and S3). The spectra of the compounds were consistent with Mn(II) and Zn(II) centers in compounds **1** and **2**, respectively [76–82]. The absorption peaks for the $\pi \rightarrow \pi^*$ transition of the aromatic ligands were obtained at the expected positions [83,84].

*2.4. Thermogravimetric Analysis*

The thermogravimetric curves of compounds **1** and **2** were obtained in the temperature range 25–1000 °C at a heating rate of 10 °C/min under $N_2$ atmosphere (Figure S4). For compound **1**, in the temperature range of 230–390 °C, the 2-ClBzH moiety present in the lattice and one coordinated phen moiety were decomposed with the observed weight loss of 47.50% (calcd. = 52.35%) [85,86]. In the temperature range 391–520 °C, the decomposition of another coordinated phen moiety with a weight loss of 29.20% (calcd. = 28%) [86] was observed. In the final step, the loss of one coordinated Cl ion in the temperature range 521–960 °C was observed with a weight loss of 7% (calcd. = 5.52%) [87]. For compound **2**, in the temperature range 180–273 °C, two coordinated 2-AmPy moieties underwent thermal decomposition with an observed weight loss of 37.66% (calcd. = 35.88%) [88]. One coordinated 4-MeBz moiety and –$CO_2$ and –$CH_3$ fragments from the other coordinated 4-MeBz moiety underwent decomposition in the temperature range 274–650 °C with an observed weight loss of 37.64% (calcd. 37.02%) [89].

### 2.5. Theoretical Study

This theoretical investigation deals with the enclathration of the H-bonded 2-ClBzH dimer within the supramolecular host cavity formed by complex molecules viz. [Mn(phen)$_2$Cl$_2$]. This study commenced with the calculation of the molecular electrostatic potential (MEP) surfaces for both [Mn(phen)$_2$Cl$_2$] and 2-ClBzH co-formers to pinpoint their nucleophilic and electrophilic regions. The MEP analysis of [Mn(phen)$_2$Cl$_2$] (Figure 11a) revealed the presence of nucleophilic zones at the chloride moieties (−72.5 kcal/mol) and electrophilic zones at the phen with the MEP maximum on the aromatic hydrogen atoms (+28.2 kcal/mol). This polarization correlates with the formation of a 2D-layered structure along the *ab* crystallographic plane, dominated by CH⋯Cl interactions, as shown in Figure 6. The MEP surface for the 2-ClBzH molecule (Figure 11b) displays an expected MEP maximum at the acidic hydrogen (+50 kcal/mol) and a minimum at the oxygen atom (−34.5 kcal/mol), with negative potentials also observed at the chlorine belt (−12.5 kcal/mol) and over the aromatic ring's center (−2.5 kcal/mol). Upon dimerization, the MEP landscape of 2-ClBzH transformed, with the maximum over the aromatic hydrogen atoms (20.0 kcal/mol) and the minimum at the region influenced by oxygen and chlorine atoms (−28.2 kcal/mol, see Figure 11c), highlighting an electron-rich surface extending from the core of four oxygen atoms to the chlorine's belts and the π-basic aromatic rings, revealing a pronounced complementarity with the positive cleft of [Mn(phen)$_2$Cl$_2$].

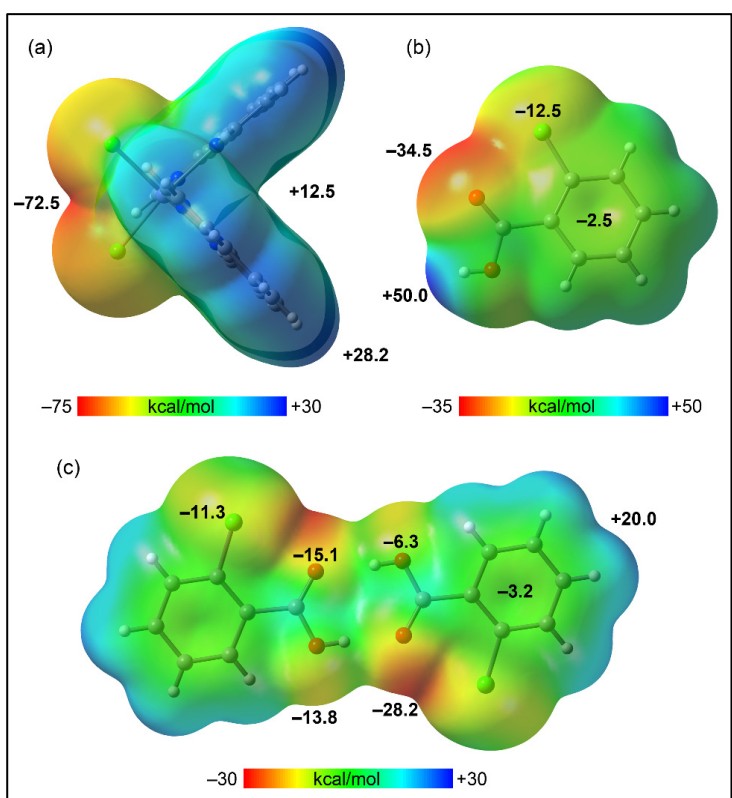

**Figure 11.** MEP surfaces of [Mn(phen)$_2$Cl$_2$] (**a**), 2-ClBzH (**b**) and its dimer (**c**), setting an isovalue of 0.001 a.u.

Figure 12a presents the QTAIM and NCI plot analysis for the 2-ClBzH dimer, showcasing bond critical points (BCPs), bond paths, and blue RDG isosurfaces [(signλ$_2$)×ρ ≈ −0.035 a.u.] for each OH⋯O bond, evidencing the strong nature of the H-bonds with a significant dimerization energy of −13.7 kcal/mol. This robustness explains the dimers' formation in the solid state. Further analysis of the dimer's interaction with two [Mn(phen)$_2$Cl$_2$] molecules to create a tetrameric assembly revealed a substantial interaction energy of −47.5 kcal/mol, driven by multiple cooperative interactions, predominantly analyzed

through NCI plot for clarity. The interactions included CH···Cl and Cl···π(phen), highlighted by green RDG isosurfaces. Moreover, larger RDG isosurfaceswere also observed between the oxygen atoms and the π-systems of phen ligands, indicating the formation of O···π interactions in a T-shaped arrangement (outlined in Figure 12b using dashed rectangles). Additionally, extensive RDG isosurfaces above and below the 2-ClBzH dimer characterized the electrostatically enhanced π-stacking interactions that embraced two phen ligands (above and below the dimer) and the entire 2-ClBzH dimer, including the aromatic rings and the supramolecular $R_2^2(8)$ ring. In Cl···π, O···π and π···π van der Waals interactions are attractive, as revealed by the sign of $\lambda_2$, which is negative in the RDG surfaces that characterize these interactions. The (sign$\lambda_2$)x$\rho$ varies from −0.015 to 0.005 a.u. This complex interplay of forces explains the significant formation of energy and suggests that the formation of the H-bonded dimer notably increases its potential for interaction with [Mn(phen)$_2$Cl$_2$], promoting its enclathration.

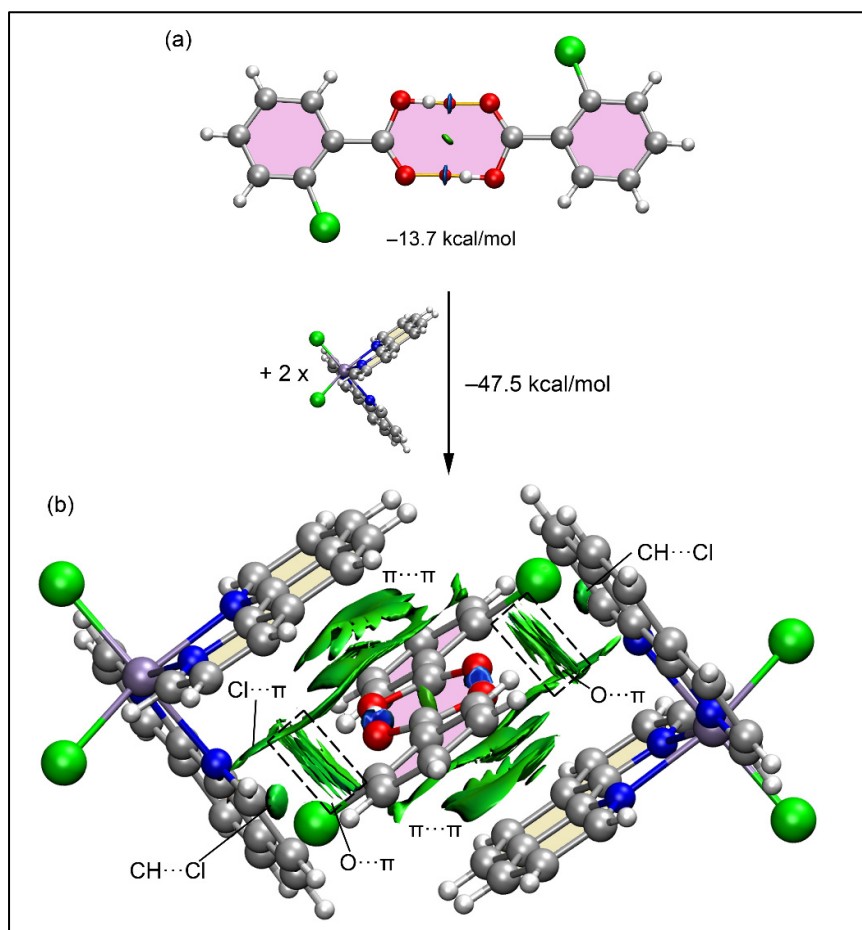

**Figure 12.** (**a**) QTAIM and NCI plot analysis of the self-assembled dimer of 2-ClBzH and the dimerization energy. (**b**) NCI plot analysis of the tetrameric assembly and the formation energy starting from the dimer and two molecules of [Mn(phen)$_2$Cl$_2$]. Only intermolecular interactions are shown.

## 3. Materials and Methods

All chemicals viz. manganese(II) chloride tetrahydrate, anhydrous zinc(II) chloride, 2-aminopyridine, 2-chlorobenzoic acid, 4-methylbenzoic acid, and 1,10-phenanthroline used for synthesis were obtained from commercial sources and used as received. Elemental analyses (C, H, and N) were carried out using the Perkin Elmer 2400 series II CHNS/O analyzer. IR spectra included KBr pellets with a Bruker Alpha (II) infrared spectrophotometer from 4000 to 500 cm$^{-1}$. The diffuse-reflectance UV-Vis spectra of solid powder samples **1**

and **2** were obtained using a Shimadzu UV-2600 spectrophotometer. After collecting the data by directing light onto the sample and measuring the amount of scattered light, it was converted to the Kubelka–Munk function, which analyzed the data and transformed them into absorption spectra. Mathematical equations were used in this transformation to establish a relationship between the sample's light absorption and the K-M function [90]. BaSO4 powder was employed as a reference to establish 100% reflectance for solid-state UV-Vis NIR spectra. Room temperature magnetic susceptibilities were assessed at 300 K using the Evans method on the Sherwood Mark 1 Magnetic Susceptibility Balance. Thermogravimetric analyses were carried out in the 25–500 °C range (at the heating rate of 10 °C min$^{-1}$) under a N$_2$ atmosphere on a Mettler Toledo TGA/DSC1 STAR$^e$ system.

### 3.1. Syntheses

#### 3.1.1. Synthesis of [Mn(phen)$_2$Cl$_2$]2-ClBzH (**1**)

The Mn(II) complex was prepared by dissolving 0.360 g (2 mmol) of Phen in 10 mL of deionized water in a round bottom flask, to which an aqueous solution (5 mL) of 0.197 g (1 mmol) of MnCl$_2$·4H$_2$O was added with continuous stirring before being left at room temperature for about an hour. To the resulting solution, an aqueous solution (5 mL) of 0.156 g (1 mmol) of 2-chlorobenzoic acid was added slowly, and the mixture was kept under mechanical stirring for another hour (Scheme 1). The resultant solution was left undisturbed in cooling conditions (2–4 °C); yellow block-shaped crystals were obtained after a few days. Yield: 0.520 g (81.12%). Anal. calcd. For C$_{31}$H$_{21}$Cl$_3$MnN$_4$O$_2$, C, 57.92%; H, 3.29%; N, 8.72%. The following was found: C, 57.82%; H, 3.18%; and N, 8.60%. FT-IR (KBr pellet, cm$^{-1}$): 3411(br), 3063(w), 2668(w), 2550(w), 1690(s), 1605(m), 1524(s), 1420(s), 1334(m), 1296(s), 1151(w), 1123(sh), 1105(w), 931(w), 857(w), 791(w), and 716(w) (s, strong; m, medium; w, weak; br, broad; sh, shoulder).

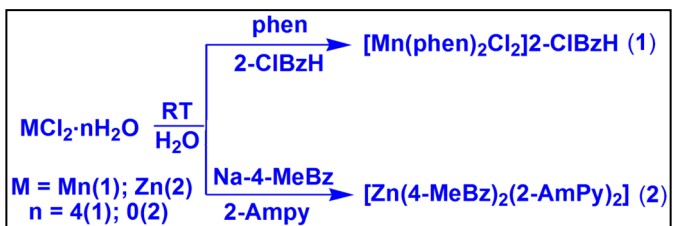

**Scheme 1.** Syntheses of compounds **1** and **2**.

#### 3.1.2. Synthesis of [Zn(4-MeBz)$_2$(2-AmPy)$_2$] (**2**)

Zinc(II) chloride at 0.136 g (1 mmol) was dissolved in 5 mL of deionized water, to which an aqueous solution (5 mL) of 0.316 g (2 mmol) of sodium salt and 4-methylbenzoic acid was added drop by drop with continuous stirring for an hour. After 1 h, an aqueous solution (5 mL) of 0.188 g (2 mmol) of 2-AmPy was added to the solution and left to stir for another hour (Scheme 1). The resulting solution was left undisturbed in a refrigerator below 4 °C for crystallization; from this, colorless prism-shaped crystals suitable for single-crystal X-ray diffraction were obtained after a few days. Yield: 0.468 g (89.56%). Anal. calcd. For C$_{26}$H$_{26}$N$_4$O$_4$Zn, C, 59.61%; H, 5.00%; N, 10.69%. The following was found: C, 59.55%; H, 4.95%; N, 10.60%. FT-IR (KBr pellet, cm$^{-1}$): 3338(br), 2729(w), 2925(w), 1645(s), 1610(s), 1566(s), 1496(s), 1453(s), 1439(m), 1272(m), 1163(w), 1151(w), 1096(w), 1009(w), 860(w), 790(w), 770(w), 741(w), 695(w), and 666(m) (s, strong; m, medium; w, weak; br, broad; sh, shoulder).

### 3.2. Crystallographic Data Collection and Refinement

The single crystal XRD data of the collections for compounds **1** and **2** were acquired employing a Bruker D8 Venture diffractometer (Karlsruhe, Germany) equipped with a Photon III 14 detector and utilizing an Incoatec high brilliance IμS DIAMOND tube [Cu/Kα radiation (λ = 1.54178 Å)], along with Incoatec Helios MX multilayer optics. The data

collection was performed at 100 K for the crystals. Scaling and absorption corrections were performed using the SADABS program for all datasets [91]. Crystal structures were solved by a direct method and refined on $F^2$ by a full matrix least squares technique with SHELXL-2018/3 [92] using the WinGX [93] platforms. Non-hydrogen atoms were refined with anisotropic thermal parameters. The hydrogen atoms of the organic ligands were placed in ideal positions and refined as riding atoms. Diamond 3.2 software was used for graphical illustrations [94]. The crystallographic data of compounds **1** and **2** are summarized in Table 3, and CCDC deposition numbers are cited in Appendix A.

**Table 3.** Crystallographic data and structure refinement details for **1** and **2**.

| Parameters | 1 | 2 |
|---|---|---|
| Formula | $C_{31}H_{21}Cl_3MnN_4O_2$ | $C_{26}H_{26}N_4O_4Zn$ |
| Formula weight | 642.81 | 523.88 |
| Temp, [K] | 100 | 100 |
| Crystal system | Triclinic | Monoclinic |
| Space group | $P\bar{1}$ | Cc |
| a, [Å] | 10.6563(18) | 9.9347(14) |
| b, [Å] | 10.9066(19) | 23.521(3) |
| c, [Å] | 12.790(2) | 10.5889(15) |
| α, [°] | 89.159(9) | 90 |
| β, [°] | 66.391(7) | 93.195(4) |
| γ, [°] | 86.483(8) | 90 |
| V, [Å$^3$] | 1359.4(4) | 2470.5(6) |
| Z | 2 | 4 |
| Absorption coefficient (mm$^{-1}$) | 6.977 | 1.709 |
| F(0 0 0) | 654.0 | 1088.0 |
| $\rho_{calc}$g/cm$^3$ | 1.570 | 1.409 |
| index ranges | $-12 \leq h \leq 12$, $-12 \leq k \leq 13$, $-15 \leq l \leq 15$ | $-11 \leq h \leq 11$, $-28 \leq k \leq 28$, $-12 \leq l \leq 12$ |
| Crystal size, [mm$^3$] | $0.38 \times 0.28 \times 0.25$ | $0.38 \times 0.31 \times 0.15$ |
| 2θ range, [°] | 8.122 to 137.828 | 9.676 to 136.904 |
| Independent Reflections | 4832 | 4216 |
| Reflections collected | 53,812 | 26,679 |
| Refinement method | Fullmatrix Leastsquares for F$^2$ | Fullmatrix Leastsquares for F$^2$ |
| Data/restraints/parameters | 4832/0/371 | 4216/2/319 |
| Goodness-of-fit on F$^2$ | 1.060 | 0.831 |
| Final R indices [I > 2σ(I)] | R$_1$ = 0.0773, wR$_2$ = 0.2158 | R$_1$ = 0.0281, wR$_2$ = 0.0725 |
| R indices (all data) | R$_1$ = 0.0812, wR$_2$ = 0.2233 | R$_1$ = 0.0281, wR$_2$ = 0.0725 |
| Largest hole and peak [e·Å$^{-3}$] | 1.00/−1.06 | 0.73/−0.37 |

### 3.3. Computational Methods

Single-point calculations were conducted using the Turbomole 7.7 program [95] at the RI-BP86-D4/def2-TZVP level of theory [96–98]. In our study, we employed computational methods that utilized X-ray-determined coordinates for all atoms except hydrogen. As mentioned, this approach involves optimizing only the position of hydrogen atoms while retaining the fixed positions of heavier atoms as determined experimentally. This methodology is commonly adopted in the analysis of non-covalent interactions within the solid state to maintain the integrity of the molecular framework, as observed in the crystal structure. The purpose of this approach is to focus on analyzing the strength and characteristics of existing contacts within the solid-state context rather than exploring potential configurations that might be more favorable in an isolated gas-phase environment. To analyze these interactions, Bader's "Atoms in molecules" theory (QTAIM) [99] and the non-covalent interaction plot (NCI plot) [100] were employed via the Multiwfn program [101], with visualizations generated using VMD visualization software version 1.9 [102]. The binding energies were calculated using a supramolecular approach, subtracting the sum of the

energies of the monomers from the energy of the assembled complex. The molecular electrostatic potential (MEP) surface was represented at an isosurface of 0.001 a.u., reflecting the van der Waals surface.

## 4. Conclusions

Two new Mn(II) and Zn(II) metal–organic compounds involving the 1,10-phenanthroline and methyl-benzoate were synthesized and characterized using single crystal X-ray diffraction, electronic, FT-IR, and TGA analyses. The crystal structure analysis of compound **1** revealed the dimerization of 2-ClBzH moieties present in the lattice and their subsequent enclathration within the hexameric supramolecular host cavity formed by orderly monomeric units. Similarly, the crystal structure analysis of compound **2** unfolded the dual mode of coordination of 4-CH$_3$Bz with the metal center and their role in the self-aggregation of the individual units towards the formation of novel supramolecular architectures. Moreover, non-covalent interactions involving lp(O)-$\pi$, lp(Cl)-$\pi$, C–H$\cdots$Cl, and $\pi$-stacking interactions as well as N–H$\cdots$O, C–H$\cdots$O and C–H$\cdots\pi$ hydrogen bonding interactions were found to be involved in stabilizing the molecular self-association of the compounds. The theoretical investigation provides some insights into the mechanism of supramolecular assembly in compound **1**. The MEP surface analysis of the 2-ClBzH dimer in **1** reveals the existence of an electron-rich surface, encompassing the oxygen, chlorine, and $\pi$-basic atoms of the aromatic ring and suggesting symbiosis with the positive cleft of [Mn(phen)$_2$Cl$_2$]. Further energetically significant dimerization energy of 2-ClBzH and substantial interaction energy of the dimer with [Mn(phen)$_2$Cl$_2$] molecules suggest the formation of the dimer, its enclathration within the hexameric host, and the cooperative nature of multiple non-covalent interactions. These findings enrich our understanding of the principles governing the design and stabilization of complex supramolecular structures, potentially guiding future research and applications in materials science and molecular engineering.

**Supplementary Materials:** The following supporting information can be downloaded at https://www.mdpi.com/article/10.3390/inorganics12050139/s1. Table S1: Selected bond lengths (Å) and bond angles (°) of Ni(II) and Co(II) centers in compounds **1** and **2**, respectively. Figure S1: FT-IR spectra of compounds **1** and **2**; Figure S2: (a) UV-Vis-NIR spectrum of **1**, (b) UV-Vis spectrum of **1**; Figure S3: (a) UV-Vis-NIR spectrum of **2**, (b) UV-Vis spectrum of **2**; Figure S4: Thermogravimetric curves of the compounds **1** and **2**.

**Author Contributions:** Conceptualization, A.F. and M.K.B.; methodology, A.F. and M.K.B.; software, A.F. and R.M.G.; formal analysis, A.F.; investigation, M.B., S.B. and R.M.G.; data curation, M.B.-O.; writing—original draft preparation, M.B.; writing—review and editing, M.K.B.; visualization, A.F.; supervision, M.K.B.; project administration, A.F. and M.K.B.; funding acquisition, A.F. and M.K.B. All authors have read and agreed to the published version of the manuscript.

**Funding:** Financial support was provided by SERB-SURE (Grant number: SUR/2022/001262); ASTEC, DST, Govt. of Assam (grant number ASTEC/S&T/192(177)/2020-2021/43); and the Gobierno de Espana, MICIU/AEI (project number PID2020-115637GB-I00), which we gratefully acknowledged. The authors thank IIT-Guwahati for the TG data.

**Data Availability Statement:** Data are contained within the article and Supplementary Materials.

**Conflicts of Interest:** The authors declare no conflicts of interest. The funders had no role in the design of the study; in the collection, analyses, or interpretation of data; in the writing of the manuscript; or in the decision to publish the results.

## Appendix A

CCDC 2322621and 2322622 contain the supplementary crystallographic data for compounds **1** and **2**. These data can be obtained free of charge at http://www.ccdc.cam.ac.uk or from the Cambridge Crystallographic Data Centre, 12 Union Road, Cambridge CB2 1EZ, UK; fax: (+44) 1223-336-033; or E-mail: deposit@ccdc.cam.ac.uk.

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
