# Peer review of "Supramolecular Assemblies in Mn(II) and Zn(II) Metal–Organic Compounds Involving Phenanthroline and Benzoate: Experimental and Theoretical Studies"

_inorganics, doi:10.3390/inorganics12050139_

Round 1

Reviewer 1 Report

Comments and Suggestions for Authors

The manuscript describes two different metal-organic frameworks with different metals and ligands from the crystallographic analysis. It is interesting to see that different types of molecular interactions lead to three-dimensional structures. However, they have no common metals or ligands as well as their molecular interactions and I wonder why the authors treated them in one manuscript.

The authors measured the FT-IR spectra of two compounds and absorption bands are listed in 3.1. Each compound has two ligand species and their absorption bands may overlap and shift from those of the isolated compounds, which would make their assignments complicated. I wonder how the authors could make assignments of the absorption bands of the compounds as was described in 2.3.1. In the last part of this section, the authors assigned two bands in both compounds to the vibrational modes of coordinating phenanthrene. This would be corrected as the compound 2 do not contain phenanthrene.

The electronic spectra of the two compounds are given in Figures S2 and S3 in the supplementary materials in the solid and aqueous state. However, Chapter 3 of Materials and Methods describes that the UV-Vis spectra were obtained by the diffuse reflectance method, which would only be applied to the solid powder sample. The method employed to transform the diffuse reflectance spectra into the absorption spectra should also be described.

It is understandable that due to their electronic configurations both Mn(II) and Zn(II) do not give any absorption bands in the UV-Vis spectra. This may contradict the description in the manuscript that the spectra reveal the presence of the metals in the compounds.

The peak positions of Figures S2 and S3 are in good agreement with those of 1,10-phenanthroline. While it is understandable that the spectra of compound 1 are dominated by the absorption of phen, it is difficult to understand that the spectra of compound 2 in Figure S3 are very similar to those in Figure S2, as phen is absent in compound 2. 

Author Response

First we would like to thank the referee for his/her careful reading of the manuscript, corrections and suggestions, our responsess follow:

Comments and Suggestions for Authors

Q1. The manuscript describes two different metal-organic frameworks with different metals and ligands from the crystallographic analysis. It is interesting to see that different types of molecular interactions lead to three-dimensional structures. However, they have no common metals or ligands as well as their molecular interactions and I wonder why the authors treated them in one manuscript.

Reply: Although we have described two different metal-organic compounds with different metals and ligands, and having no common molecular interactions yet the presence of benzoic acid derivatives i.e. chlorobenzoic acid and methyl benzoate in the reported Mn(II) and Zn(II) metal-organic compounds, respectively justifies that they can be reported in the same manuscript.

Q2. The authors measured the FT-IR spectra of two compounds and absorption bands are listed in 3.1. Each compound has two ligand species and their absorption bands may overlap and shift from those of the isolated compounds, which would make their assignments complicated. I wonder how the authors could make assignments of the absorption bands of the compounds as was described in 2.3.1.

 Reply: The assignment of the absorption bands in section 2.3.1 has been done after prior literature review. We have found that the specified ligands (2-ClBzH, 2-AmPy, 4-MeBz and 1,10-phenanthroline) show shift of absorption bands in the regions of higher or lower wavenumbers in presence of auxillary ligands. Therefore, specific references of compounds with similar observations have been cited for justification of the assignment of IR spectra of the compounds.

Q3. In the last part of this section, the authors assigned two bands in both compounds to the vibrational modes of coordinating phenanthrene. This would be corrected as the compound 2 do not contain phenanthrene.

Reply: We have now corrected it in the revised manuscript. 

Q4. The electronic spectra of the two compounds are given in Figures S2 and S3 in the supplementary materials in the solid and aqueous state. However, Chapter 3 of Materials and Methods describes that the UV-Vis spectra were obtained by the diffuse reflectance method, which would only be applied to the solid powder sample. The method employed to transform the diffuse reflectance spectra into the absorption spectra should also be described.

Reply: The methodology part has been revised as suggested by the esteemed reviewer. We have now described the method employed to transform the diffuse reflectance spectra into the absorption spectra in the Materials and Methods section in the revised manuscript with proper reference citation.

Q5. It is understandable that due to their electronic configurations both Mn(II) and Zn(II) do not give any absorption bands in the UV-Vis spectra. This may contradict the description in the manuscript that the spectra reveal the presence of the metals in the compounds.

eply: We have now used the term “consistent with” in lieu of “reveals the presence of” in the section 2.3.2 while referring to the observed UV-Vis spectra and stating them to have been obtained for Mn(II) and Zn(II) ions present in the compounds. Similar spectra have been reported for Mn(II) and Zn(II) compounds with similar ligands in the literature and we have cited a few of them in the manuscript. Further, the magnetic moment values found for our compounds experimentally, also support the presence of Mn(II) and Zn(II) metal centers in our reported compounds.

Q6. The peak positions of Figures S2 and S3 are in good agreement with those of 1,10-phenanthroline. While it is understandable that the spectra of compound 1 are dominated by the absorption of phen, it is difficult to understand that the spectra of compound 2 in Figure S3 are very similar to those in Figure S2, as phen is absent in compound 2.

Reply: Although the spectra of compound 2 in figure S3 is similar to that for compound 1 in figure S2 dominated by phen, yet we have cited and found adequate references to support our statement that the absorption bands obtained in the UV region for compound 2 in figure S3 have been due to π→π* transitions of the aromatic ligands (particularly 2-Ampy) [Polyhedron 2009, 28, 3526–3532, doi:10.1016/j.poly.2009.05.079; International Journal of Biomaterials 2021, 2021, e4981367, doi:10.1155/2021/4981367].

Reviewer 2 Report

Comments and Suggestions for Authors

I reviewed the manuscript ID inorganics-2989726 entitled “Supramolecular Assemblies in Mn(II) and Zn(II) Metal-organic Compounds involving Phenanthroline and Benzoate: Experimental and Theoretical Studies”.  The manuscript reports the synthesis, spectroscopic studies, X-ray structures, and computational studies of one of the two new systems reported.  While the quality of the manuscript is high, I do not understand how the Zn complex and Mn complex are related except that they share in common a benzoic acid derivative, though in the Zn complex the benzoic acid derivative is a ligand and is not a ligand in the Mn complex.  The addition of the Zn complex to the more complete story about the Mn complex diminishes the study of the Mn complex.  I don’t understand the rationale of not completing some sort of computational studies of the Zn complex for some sort of comparison to the Mn complex.  Unless a more complete study of 2 is included in a revised manuscript, removing the Zn complex seems needed.   

The introduction provides significant reasons for the motivation for choice of these systems to study and related work but not how these two systems are related to each other. The synthetic preparations include sufficient detail for reproduction, and are fully characterized vis IR, UV-Vis, TGA, elemental analyses, and X-ray structures.  The spectroscopic data is mostly interpreted properly. The X-ray structures are analyzed thoroughly including the significant intra- and intermolecular features.    Numerous comparisons to previous related work are made throughout the manuscript.  Numerous, appropriate, and modern references are included.  Figures, ESI, and Tables are formatted well.  While the quality of the work is high, the connection between the systems studies is not clear at all.  If the Zn complex is entirely removed OR a theoretical study of some sort is carried out on the Zn complex and explicitly compared to the Mn, then I could recommend accepting the manuscript with major revisions.

In addition, some important edits/changes/questions need to be addressed before a final version of the manuscript should be accepted for publication.

1. Why are so many reflections missing from the Zn structure data collection?

2.  Why is the data:parameter ratio so low (around 7) for the Mn structure?

3.  p2, line 69, replace the two uses of “it” with helpful words

4.  everywhere in the manuscript where angles/distances are included, include esds if they involve refined atom positions (bond lengths, angles between planes, D-A H bond distances, etc.)

5.  Table 1 says nothing important – put in the ESI

6.  wherever planes within the unit cell are noted, make sure they are italicized

7.  figure 9 is so large the intermolecular interactions are too hard to see: include fewer molecules so the figure can be zoomed in further for details to show up well

8.  section 2.3.2:  the UV-Vis spectra DO NOT reveal the presence of Mn(II) and Zn(II) as the absence of  any d-d bands (Zn) or spin forbidden low epsilon d-d bands (Mn) is not positive evidence, this is negative evidence.  Indicate “consistent with” or remove the interpretation altogether

9.  theoretical study: where the X-ray coordinates used for the theoretical studies?  I assume so but this needs to be said explicitly in the experimental section

10.  fig 12 and discussion: can you explicitly indicate whether the green surfaces indicate vdW interactions, if they are attractive/repulsive, and max values of the interactions as determined form the various surface regions (O…pi, pi…pi, Cl…pi) as a way to evaluate what the authors indicate are the strongest contributors to these dimers

11.  the last few refs are missing titles and have incorrect formatting

Comments on the Quality of English Language

very few edits needed realted to use of Enlgish language

Author Response

First, we would like to thank this referee for his/her careful reading of the manuscript, corrections and suggestions. Our responses follow:

Q1. I reviewed the manuscript ID inorganics-2989726 entitled “Supramolecular Assemblies in Mn(II) and Zn(II) Metal-organic Compounds involving Phenanthroline and Benzoate: Experimental and Theoretical Studies”.  The manuscript reports the synthesis, spectroscopic studies, X-ray structures, and computational studies of one of the two new systems reported.  While the quality of the manuscript is high, I do not understand how the Zn complex and Mn complex are related except that they share in common a benzoic acid derivative, though in the Zn complex the benzoic acid derivative is a ligand and is not a ligand in the Mn complex.  The addition of the Zn complex to the more complete story about the Mn complex diminishes the study of the Mn complex.

Reply: Our report addresses the molecular self assembly. The behavior of common benzoic acid derivatives in both the compounds is different and that makes the molecular association different and interesting in the compounds.

Q2.  I don’t understand the rationale of not completing some sort of computational studies of the Zn complex for some sort of comparison to the Mn complex.  Unless a more complete study of 2 is included in a revised manuscript, removing the Zn complex seems needed.

Reply: Thank you for your comments and suggestions regarding the comparative analysis of the Zn and Mn complexes in our manuscript. We understand the reviewer's interest in a comprehensive computational study to compare these two complexes. However, due to the tight revision timeframe of 10 days set by the editorial office, it is not feasible for us to conduct a detailed theoretical study on the Zn complex at this stage. Regarding the inclusion of the Zn complex (compound 2) in the manuscript, we wish to clarify that its inclusion is justified on the grounds of its novelty and the completion of its spectroscopic characterization and structural elucidation via X-ray crystallography. While we acknowledge that a computational study would enrich the understanding of its properties and potentially validate the experimental findings, the absence of such a study does not detract from the value of the experimental data presented. The primary focus of our manuscript is on the experimental findings related to these newly synthesized complexes. Removing the experimental data concerning the Zn complex solely due to the lack of corresponding computational analysis would omit valuable information about a novel compound, which has been characterized by established experimental techniques. Therefore, we propose to maintain the section on the Zn complex in the manuscript, emphasizing the experimental insights it provides.

The introduction provides significant reasons for the motivation for choice of these systems to study and related work but not how these two systems are related to each other. The synthetic preparations include sufficient detail for reproduction, and are fully characterized vis IR, UV-Vis, TGA, elemental analyses, and X-ray structures.  The spectroscopic data is mostly interpreted properly. The X-ray structures are analyzed thoroughly including the significant intra- and intermolecular features.    Numerous comparisons to previous related work are made throughout the manuscript.  Numerous, appropriate, and modern references are included.  Figures, ESI, and Tables are formatted well.

  Q3. While the quality of the work is high, the connection between the systems studies is not clear at all. 

 Reply:  Although we have described two different metal-organic compounds with different metals and ligands, and having no common molecular interactions yet the presence of benzoic acid derivatives i.e. chlorobenzoic acid and methyl benzoate in the reported Mn(II) and Zn(II) metal-organic compounds, respectively justifies that they can be reported in the same manuscript. Also, the different nature of the benzoic acid derivatives in the compounds makes molecular association interesting.

 Q4. If the Zn complex is entirely removed OR a theoretical study of some sort is carried out on the Zn complex and explicitly compared to the Mn, then I could recommend accepting the manuscript with major revisions.

Reply: See or response to Q2 above

Q5. In addition, some important edits/changes/questions need to be addressed before a final version of the manuscript should be accepted for publication.

  1. Why are so many reflections missing from the Zn structure data collection?

Reply: The Zn structure was found to be twinned [twin law (-1, 0, 0, 0, -1, 0, 0, 0, -1) and BASF 0.13(2)]. During the refinement 8 reflections were OMITTED, as you can see in the .res file, due to their high associated error and, probably, the other missing reflections are due to the beam stopper.

  1. 2. Why is the data:parameter ratio so low (around 7) for the Mn structure?

Reply: Mn structure has a good data: parameter ratio. We can again refer to the Zn structure. As already said, being a twinned structure usually needs a higher amount of data. Unfortunately, in that case the crystals were already measured and it was not possible to remount it, due to X-ray damage.

Nevertheless, CHECKCIF found these problems as level C, not A or B, so we considered at that time that it would not suppose a major problem.

  1. p2, line 69, replace the two uses of “it” with helpful words

Reply: We have now replaced the use of “it” with some other helpful words in the revised manuscript.

  1. everywhere in the manuscript where angles/distances are included, include esds if they involve refined atom positions (bond lengths, angles between planes, D-A H bond distances, etc.)

 Reply: We have included the esds in the bond lengths, angles between planes, D-A H bond distances, etc. in the revised manuscript.

  1. Table 1 says nothing important – put in the ESI

Reply: We have moved Table 1 in the ESI.

  1. wherever planes within the unit cell are noted, make sure they are italicized

Reply: We have indicated the planes in italics in our revised manuscript.

  1. figure 9 is so large the intermolecular interactions are too hard to see: include fewer molecules so the figure can be zoomed in further for details to show up well

 Reply: We have now included fewer molecules in figure 9 so that it becomes easy to zoom in and look up for details in the revised manuscript.

  1. section 2.3.2: the UV-Vis spectra DO NOT reveal the presence of Mn(II) and Zn(II) as the absence of any d-d bands (Zn) or spin forbidden low epsilon d-d bands (Mn) is not positive evidence, this is negative evidence.  Indicate “consistent with” or remove the interpretation altogether

Reply: We have now used the term “consistent with” in the section 2.3.2 while referring to the observed UV spectra and stating them to have been obtained for Mn(II) and Zn(II) ions. We have cited and found numerous references that are well consistent with our observations. In addition, the magnetic moment values found for our compounds experimentally, also support the presence of Mn(II) and Zn(II) metal centers in our reported compounds.

  1. theoretical study: where the X-ray coordinates used for the theoretical studies? I assume so but this needs to be said explicitly in the experimental section

Reply: Thank you, a statement has been added

  1. fig 12 and discussion: can you explicitly indicate whether the green surfaces indicate vdW interactions, if they are attractive/repulsive, and max values of the interactions as determined form the various surface regions (O…pi, pi…pi, Cl…pi) as a way to evaluate what the authors indicate are the strongest contributors to these dimmers

Reply: This has been done when commenting the results of Fig. 12

 the last few refs are missing titles and have incorrect formatting

Reply: We have added the titles and corrected the formatting in the last few references in the revised manuscript.

Round 2

Reviewer 2 Report

Comments and Suggestions for Authors

While I respectfully diagree with the authors explanation of how the two complexes are related, leaving the Zn complex in the manuscript is fine.  The others changes to the manuscript and responses to my orginal concerns are sufficient for accpetance of the manuscript in the current revised form.